# A Point Mutation in Cassette Relieves the Repression Regulation of CcpA Resulting in an Increase in the Degradation of 2,3-Butanediol in *Lactococcus lactis*

**DOI:** 10.3390/microorganisms12040773

**Published:** 2024-04-11

**Authors:** Xian Xu, Fulu Liu, Wanjin Qiao, Yujie Dong, Huan Yang, Fengming Liu, Haijin Xu, Mingqiang Qiao

**Affiliations:** 1School of Life Science, Shanxi University, Taiyuan 030006, China; m17835425141@163.com (X.X.); 18434773155@163.com (Y.D.); 15877311433@163.com (H.Y.); lfm@sxu.edu.cn (F.L.); 2State Key Laboratory of Coordination Chemistry, Chemistry and Biomedicine Innovation Center, School of Chemistry and Chemical Engineering, Nanjing University, Nanjing 210023, China; 1120170379@mail.nankai.edu.cn; 3The Key Laboratory of Molecular Microbiology and Technology, Ministry of Education, College of Life Sciences, Nankai University, Tianjin 300071, China; qiaowanjin@outlook.com (W.Q.); nkxuhaijin@163.com (H.X.)

**Keywords:** *Lactococcus lactis* N8, carbon catabolite protein A (CcpA), catabolite responsive elements (*cre*), 2,3-butanediol metabolism

## Abstract

In lactic acid bacteria, the global transcriptional regulator CcpA regulates carbon metabolism by repressing and activating the central carbon metabolism pathway, thus decreasing or increasing the yield of certain metabolites to maximize carbon flow. However, there are no reports on the deregulation of the inhibitory effects of CcpA on the metabolism of secondary metabolites. In this study, we identified a single-base mutant strain of *Lactococcus lactis* N8-2 that is capable of metabolizing 2,3-butanediol. It has been established that CcpA dissociates from the catabolite responsive element (*cre)* site due to a mutation, leading to the activation of derepression and expression of the 2,3-butanediol dehydrogenase gene cluster (*butB* and *butA*). Transcriptome analysis and quantitative polymerase chain reaction (Q-PCR) results showed significant upregulation of transcription of *butB* and *butA* compared to the unmutated strain. Furthermore, micro-scale thermophoresis experiments confirmed that CcpA did not bind to the mutated *cre*. Furthermore, in a bacterial two-plasmid fluorescent hybridization system, it was similarly confirmed that the dissociation of CcpA from *cre* eliminated the repressive effect of CcpA on downstream genes. Finally, we investigated the differing catalytic capacities of the 2,3-butanediol dehydrogenase gene cluster in *L. lactis* N8-1 and *L. lactis* N8-2 for 2,3-butanediol. This led to increased expression of *butB* and *butA*, which were deregulated by CcpA repression. This is the first report on the elimination of the deterrent effect of CcpA in lactic acid bacteria, which changes the direction of enzymatic catalysis and alters the direction of carbon metabolism. This provides new perspectives and strategies for metabolizing 2,3-butanediol using bacteria in synthetic biology.

## 1. Introduction

*Lactococcus lactis* is a group of Gram-positive bacteria with wide applications in the food industry and genetic engineering for producing recombinant proteins [1]. It has a crucial role in the dairy and health industries. *Lactococcus lactis* is generally regarded as a safe (GRAS) microorganism [1,2,3]. *Lactococcus lactis* N8 is a nisin Z producer isolated from milk in Finland, and the knowledge gained from fundamental research on this nisin production strain has been exploited for a wide variety of biotechnological applications. Because of the impressive nisin yield, research on this strain has focused on its nisin synthetic gene cluster and on increasing the production of nisin [4,5]. By analyzing phage and gene island fragments in the genome, the deletion of these fragments reduces the metabolic burden of the strain and increases the production of some substances [6]. To this end, we designed experiments to construct chassis cells by systematic cumulative deletion of prophages and transposons and some putative proteins of unknown function from the genome of the target strain *L. lactis* N8 [3] and obtained the *L. lactis* N8-1 [6] and *L. lactis* N8-2 deleted strains.

In *L. lactis*, a range of transcriptional regulatory mechanisms have been preliminarily studied. The global transcriptional regulator CcpA, a DNA-binding protein belonging to the Lacl/GalR family of transcriptional regulators [7], acts by binding to *cre* located near the promoter to repress or enhance transcription of the downstream operon [8,9]. CcpA, a regulator of genes associated with energy and nitrogen metabolism in bacteria (>300 in *Bacillus subtilis*, ≥237 in *Lactococcus lactis*) [8], acts by binding to 16-nucleotide DNA target sites known as *cre* and conducts carbon catabolite activation (CCA) and repression (CCR) processes. CcpA, which represses the transcription of genes within the tricarboxylic acid (TCA) cycle and secondary carbon source catabolism, is revealed in Gram-positive bacteria as a regulatory mechanism of carbon metabolism [10]. While the role of CcpA as a repressor of genes involved in the utilization of secondary carbon sources is well established, CcpA is also required for activation of carbon excretion pathways, including those for the production of acetate, acetoin, and glycogen, during growth in glucose [11,12,13]. The role of CcpA in regulating carbon metabolism and facing the stress response has been demonstrated in a variety of lactic acid bacteria, such as *Lactobacillus plantarum* [14], *Lactobacillus casei* [15,16], and *Lactococcus lactis* [17,18].

There is a wide range of 2,3-butanediol applications in the chemical, cosmetic, food, agriculture, and pharmaceutical industries, while 2,3-butanediol derivatives can also be used as fuel additives, polymers, and synthetic rubber [19]. 2,3-butanediol is a functional C4 compound with various industrial applications [20]. In bacteria, in addition to providing energy for cell growth, glucose metabolism produces a large number of secondary metabolites. Acetoin and 2.3-butanediol have a wide range of applications as secondary metabolites in microbial synthesis. Many microorganisms, such as *Klebsiella pneumoniae*, *Bacillus velezensis*, *Enterobacter aerogenes*, *Saccharomyces cerevisiae*, *Serratia marcescens*, *Corynebacterium glutamicum*, and *Lactococcus lactis*, have been used to efficiently produce 2,3-butanediol [21,22,23,24,25,26]. In *L. lactis*, 2,3-butanediol dehydrogenase acts as a bi-directional enzyme, allowing the interconversion of acetoin and 2,3-butanediol in organisms [27]. This conversion occurs under the conditions of NADH and NAD^+^ as coenzymes. However, little attention has been paid to 2,3-butanediol as a carbon source in central carbon metabolism for degradation.

In this study, we identified a mutant strain that could utilize 2,3-butanediol and increase its metabolism. We experimentally verified that the global transcriptional regulator CcpA deregulates the inhibitory effect of 2,3-butanediol dehydrogenase transcription by *L. lactis* N8-2. Furthermore, we elucidated the regulatory mechanism of CcpA in the metabolism of the secondary metabolite 2,3-butanediol by *L. lactis* N8-2. These findings offer new insights and a biological foundation for the role of the global transcription factor CcpA in regulating the central carbon metabolism of *Lactococcus lactis* by altering carbon flow.

## 2. Methods

### 2.1. Bacterial Strains, Plasmids, and Culture Conditions

All the strains and plasmids used in this study are described in Table 1. All the primers used are in Table 2. The growth medium employed for *L. lactis* N8-1 and *L. lactis* N8-2 was GM17, which consisted of an M17 medium supplemented with 0.5% glucose. Without agitation were incubated at 30 °C. *E. coli* DH5α and *E. coli* BL21 (DE3) strains were employed for clone screening and expression of recombinant proteins, respectively. Luria-Bertani (LB) medium, comprising 1% tryptone, 0.5% yeast extract, and 1% NaCl, was used as the growth medium. *E. coli* cultures were grown under shaking conditions at 200 rpm/min and a temperature of 37 °C, whereas BL21 (DE3) cultures were grown at a lower temperature of 16 °C to facilitate heterologous expression of recombinant proteins. GM17 and LB solid media were supplemented with 2.5% (*w*/*v*) agar powder. Clone screening was carried out using the pEASY-T1 cloning vector, which contains the *Lac*Z gene and enables white/blue color screening in plate media containing IPTG (50 mg/mL) and X-gal (20 mg/mL). The pETm3c vector was used to express the recombinant proteins. The pLEB124 vector was used as an overexpression vector in *Lactococcus lactis*, serving as a reporter vector in this study. pRSF-Dute and pACYC-eGFP plasmids were specifically engineered to enable the simultaneous expression of two target open reading frames. These plasmids were then co-transformed into *E. coli* BL21 (DE3) cells for double hybridization. To facilitate plasmid selection, the concentrations of ampicillin, kanamycin, erythromycin, and chloramphenicol used in *E. coli* were uniformly established at 50 µg/mL. The concentration of erythromycin in *Lactococcus lactis* was found to be 5 µg/mL.

### 2.2. Biolog Phenotype Microarray Metabolic Profiling

Biolog phenotype microarray microplates, which contain a tetrazolium dye that changes color because of substrate metabolism, provide a metabolic fingerprint of the microorganism [29,30]. The Phenotype microarray system (Hayward, CA, USA) was used to assess the metabolic activity of *L. lactis* N8-1 and *L. lactis* N8-2 strains using the GP2 and Gen III MicroPlate™ (Biolog Inc., Hayward, CA, USA). All experimental procedures and protocols were meticulously performed according to the manufacturer’s instructions [29]. *L. lactis* N8-1 and *L. lactis* N8-2 strains were introduced into GM17 liquid medium and cultivated until they reached the mid-exponential phase (OD_595_ = 0.6). Afterward, the bacterial solution was evenly distributed on a GM17 plate lacking resistance, and a solitary colony was selected using a sterile swab. The colonies were then inoculated into an inoculum solution. A final dilution in 20 mL of IF-C fluid (Biolog Inc., Hayward, CA, USA) was used to acquire a homogenous cell suspension without clumps [29]. The turbidity of the bacterial solution was subsequently adjusted to a 20% T value. Following this, the bacterial solution was inoculated into a microplate, and 150 μL was added to each well. The enzyme marker was set to detect a wavelength of OD_595_ nm, and the plate was incubated at a temperature of 30 °C for 18 h. Data were recorded at intervals of 6 h, and the average color change was determined by calculating the absorbance data [30]. The absorbance value of each well in the plate represents the utilization of a unique carbon source, expressed as [C-R] [31], where C and R, respectively denote the absorbance per well (measured as optical density), the absorbance of the control wells in the plate, and the number of substrates used for statistical analysis.

### 2.3. Real-Time Fluorescence Quantitative PCR

*L. lactis* N8-1 and *L. lactis* N8-2 were cultivated in GM17 medium for 6 h. The cultures were then diluted to obtain comparable cell densities. Total RNA was extracted, which was followed by reverse transcription into first-strand cDNA using the RevertAid First Strand cDNA Synthesis Kit (TransGen, Beijing, China). The gene transcription levels of *tufA*, *butA*, and *butB* were evaluated using RT-qPCR. The *tufA* [32] gene was selected as the housekeeping gene, and the comparative CT(2^−ΔΔCT^) method was used for data analysis. Transcriptions with a fold change > 2 were considered statistically significant.

### 2.4. L. lactis N8-2 Genome Complete Map Sequencing

We performed a comprehensive mapping and sequencing analysis of the entire genome of *L. lactis* N8-2, including complete splicing and assembly. The obtained sequencing results were compared with the whole-genome sequence of *L. lactis* N8 (GCA_014884605.1), available at https://www.ncbi.nlm.nih.gov/assembly/GCF_014884605.1 (accessed on 1 June 2023). This comparison enabled the identification and sorting of differential sites and sequences in *L. lactis* N8-2. To refine the screening process, the whole-genome sequence of *L. lactis* N8-2 was further validated by PCR amplification and sequencing of the amplified fragments. This approach resulted in a more accurate whole-genome sequence for *L. lactis* N8-2.

### 2.5. Detection of Promoter Activity Using Cat as a Reporter Gene

Subsequent experiments were performed using *L. lactis* N8-1 as the control strain, because *L. lactis* N8-2 is a phage-like fragment knockout based on *L. lactis* N8-1, and the P*_butBA_* promoter sequence preceding the *butBA* gene cluster in *L. lactis* N8-1 was confirmed to be free of point mutations by PCR sequencing. The laboratory reporter gene for chloramphenicol resistance, cat, was selected, and the expression vector pLEB124 was used. The genomes of *L. lactis* N8-1 and N8-2 were used as templates to amplify the original and point-mutated P*_butBA_* promoter sequences, designated as P_1_ and P_2_, respectively, using P*_butBA_*-F/R primers. Afterward, an expression plasmid was constructed by linking different promoters using plasmid pLEB124 as a template. Concurrently, plasmid pLEB124 was employed as a template to amplify the P_45_ [33] promoter using the P_45_-F/R primer. This led to the acquisition of the P_45_ promoter, which was subsequently used to construct expression plasmids with various promoters. To validate the constructs, the recombinant plasmids pLEB124-P_1_-*cat*, pLEB124-P_2_-*cat*, and pLEB124-P_45_-*cat* were obtained and individually subjected to sequencing. Afterward, the recombinant constructs were introduced into *L. lactis* N8-1 and *L. lactis* N8-2 competent cells, which resulted in the generation of the respective strains N8-1-P_45_-*cat*, N8-1-P_1_-*cat*, N8-1-P_2_-*cat*, N8-2-P_45_-*cat*, N8-2-P_1_-*cat*, and N8-2-P_2_-*cat*.

The strains obtained in the previous step were activated and transferred a second time. Cells were cultured until such time as they reached the logarithmic phase. The bacterial solution was diluted 50 times to serve as a seed solution for detection. This was followed by the preparation of a chloramphenicol-resistant LB medium with a 150 μg/mL concentration. Next, 100 μL of diluted bacterial solution was added to each well of a 96-well cell culture plate containing chloramphenicol-resistant LB medium. For each sample, three replicates were performed, and the samples were incubated at a constant temperature of 30 °C for 10 h. The P_45_ promoter was used as a positive control to determine the MIC of chloramphenicol against the bacterial strains. The expression capacity of each promoter was characterized by detecting its respective tolerance to chloramphenicol and conducting a more accurate quantitative analysis of promoter expression capacity.

### 2.6. Detection of Promoter Activity Using Red Fluorescent Protein as a Reporter Gene

The sequence of red fluorescent proteins (*rfps*) was obtained by performing *rfp*-clone-F/R cloning using plasmid pNZ8048-*rfp* as a template. Afterward, plasmids pLEB124-P_45_-*rfp*, pLEB124-P_1_-*rfp*, and pLEB124-P_2_-*rfp* were constructed by recombination to replace the cat gene in the pLEB124-P_45_-*cat*, pLEB124-P_1_-*cat*, and pLEB124-P_2_-*cat* plasmids, respectively. The newly constructed plasmids were sequenced and individually verified.

The same transformation method was employed to introduce plasmids pLEB124-P_45_-*rfp*, pLEB124-P_1_-*rfp*, and pLEB124-P_2_-*rfp* into the competent cells of *L. lactis* N8-1 and *L. lactis* N8-2. Consequently, strains N8-1-P_45_-*rfp*, N8-1-P_1_-*rfp*, N8-1-P_2_-*rfp*, N8-2-P_45_-*rfp*, N8-2-P_1_-*rfp*, and N8-2-P_2_-*rfp* were obtained. The culture was transferred to a 96-well cell culture plate in the logarithmic phase. The plates were then incubated under light-protected conditions. The fluorescence intensity of the strains was measured using ELISA after incubation for 14 h and 20 h. The parameter settings for ELISA were excitation light at 587 nm and scattering light at 610 nm. The fluorescence values obtained from each strain were used to precisely measure the promoter expression.

### 2.7. Analysis of the Functional Region Responsible for Initiating Transcription and Structural Characteristics of the butBA Gene Cluster

The genomes of *L. lactis* N8-1 and *L. lactis* N8-2 were used as templates for this study. The P*_butBA-butB_* sequence, comprising the complete P*_butBA_* and *butB* genes, was cloned using P*_butBA-butB_*-F/R primers. Additionally, the pLEB124-P_1_-*butB-cat* and pLEB124-P_2_-*butB-cat* plasmids were constructed. Afterward, these plasmids were introduced into competent cells of *L. lactis* N8-1 and *L. lactis* N8-2 through electrotransformation. This process generated N8-1-P_1_-*butB*-*cat*, N8-1-P_2_-*butB*-*cat*, N8-2-P_1_-*butB*-*cat*, and N8-2-P_2_-*butB*-*cat* strains. Resistance to chloramphenicol was assessed for each strain.

### 2.8. Preparation of a Fluorescent Probe DNA

The probes used in the MST technique comprised 5’ cy5-labeled DNA fragments. In contrast, the target fragments were obtained through two rounds of PCR amplification. In the initial amplification process, the *L. lactis* N8-1 and *L. lactis* N8-2 genomes were used as templates. PCR amplification was performed using primers with 16-bp cy5 primer sequence homology arms. The resulting amplified products were recovered by agarose gel electrophoresis after identification. Afterward, these products were constructed using pEASY-T and screened for positive clones using the white/blue color screening method. The plasmids obtained through correct sequencing were named pEASY-N8-1-*cre*, pEASY-N8-2-*cre*, and pEASY-N8-16s. In the second amplification step, the three recombinant plasmids were used as templates, with cy5 fluorescent primers acting as primers. After agarose gel electrophoresis, the PCR reaction solution was purified and recovered, leading to the formation of N8-1-*cre*, N8-2-*cre*, and N8-16s samples. These samples were then stored at −20 °C for backup. In the subsequent stages, it is essential to shield the entire procedure from exposure to light.

### 2.9. Construction of Recombinant Plasmids and Subsequent Induction of Expression for Purification of the CcpA Protein

Initially, genome information for *L. lactis* N8 was obtained from the NCBI website. The sequence information of the catabolite control protein A (*CcpA*) (Sequence ID: CP059049.1) was downloaded, and a primer design was conducted using the pETm3c vector using SnapGene software (version 6.2.0). After correct sequencing and screening of positive clones, the recombinant plasmid was chemically transformed into *E. coli* BL21 (DE3) competent cells. The plates were then subjected to overnight incubation at 37 °C, and single colonies were selected and cultured in 5-mL tubes. The expansion culture was conducted using a 1% inoculum. The culture was grown at 37 °C until the optical density at 600 nm (OD_600_) reached 0.6. Subsequently, 0.5 mM isopropyl-β-d-thiogalactopyranoside (IPTG) was added for induction for another 12 h at 16 °C. The organisms were collected by centrifugation (10,000× *g* for 15 min at 4 °C) during the crushing process. The cells were then collected and resuspended in a PB buffer solution (20 mM NaH_2_PO_4_, 20 mM Na_2_HPO_4_, and 500 mM NaCl, pH 7.4).

The pellet was resuspended in 10 mL of PB-40 buffer (20 mM NaH_2_PO_4_, 20 mM Na_2_HPO_4_, and 40 mM imidazole, pH 7.4). The cells were then disrupted by sonication. Cellular debris was eliminated by centrifugation (10,000× *g* for 15 min at 4 °C). The resulting supernatant was then filtered using a 0.22-μm membrane and loaded onto a nickel column chromatography system at 4 °C for complete binding. Afterward, heterogeneous proteins were eluted using PB-40 buffer. The target proteins were eluted using PB buffer supplemented with 0.3 M imidazole. Subsequent to elution, the proteins of interest were desalted using dextran Sephadex G-75 and collected. Finally, the target proteins were stored at −80 °C.

### 2.10. Micro-Scale Thermophoresis (MST)

We determined the binding of CcpA proteins to N8-1-*cre*, N8-2-*cre*, and N8-16s DNA fragments and also examined the affinity constants for protein CcpA target DNA binding. The N8-1-*cre*, N8-2-*cre*, and N8-16s DNA fragments labeled with the cy5 probe were initially scanned for fluorescence excitation using the PRETEST program (NanoTemper, Munich, Germany) of the micro-thermophilic kinematic instrument. This step was performed to ensure that the fluorescence intensity was within the range of 200–1000 nm. Sixteen PCR tubes were prepared for each experiment. In tubes 2 to 16, 10 μL of DNA samples at a concentration of 375 nM were added. A total of 20 μL of CcpA protein was added to the first tube at a concentration of 15 µM CcpA, and 10 μL transferred to the second tube. This two-fold dilution method was repeated for each subsequent tube, resulting in the sequential dilution of the target protein into 16 gradients. Specific quantities of the protein and DNA mixture were extracted using a capillary pipette and then deposited onto a stationary plate designed for capillary pipettes to measure the change in fluorescence distribution upon heating as a function of the concentration of the DNA-protein complex. Since the migration of individual molecules is different from that of ligand-bound molecules, the change in fluorescence distribution was used to determine the ratio of free proteins to proteins bound to the DNA. The collected data were then saved and analyzed.

### 2.11. Construction of a Bacterial Two-Plasmid Fluorescence Hybridization System

N8-1-*cre* and N8-2-*cre* DNA sequences were inserted at the N-terminus of the enhanced green fluorescent protein (eGFP) gene in the pACYC-eGFP vector, leading to the creation of pACYC-N8-1-*cre*-eGFP and pACYC-N8-2-*cre*-eGFP constructs. Furthermore, *CcpA* was inserted into the pRSF-Duet vector, resulting in the creation of the pRSF-*CcpA* construct. After validating the correct sequencing, the pRSF-*CcpA*, pACYC-N8-1-*cre*-eGFP, and pACYC-N8-2-*cre*-eGFP constructs were introduced into *E. coli* BL21(DE3) competent cells by electroporation at a voltage of 2.5 kV. The transformed bacterial cells were then plated on LB solid medium containing 50 μg/mL chloramphenicol and 50 μg/mL kanamycin and incubated overnight at 37 °C. Single colonies were selected for colony PCR validation, and plasmids confirmed to have the correct sequences were labeled as CcpA-N8-1-*cre*-eGFP and CcpA-N8-2-*cre*-eGFP. These plasmids were stored at −20 °C.

### 2.12. Determination of Growth Trends and Fluorescence Values in a Two-Plasmid Fluorescent Hybridization Assay

The four strains were activated in LB medium. The culture was grown at 37 °C until the optical density at 600 nm (OD_600_) was 0.6. Afterward, 0.5 mM IPTG was added for induction for another 12 h at 16 °C. The bacterial liquid was collected by centrifugation (9000× *g*, 4 °C, 3 min) and washed twice with PBS. The OD_600_ value was measured, and the samples were adjusted until the OD values were the same. Finally, the fluorescence intensity of the samples was determined using a fluorescence spectrophotometer at an excitation wavelength of 480 nm and an emission wavelength of 520 nm.

### 2.13. Determination of the Metabolism of 2,3-Butanediol at Different Concentrations

2,3,5-triphenyl-2H-tetrazolium chloride (TTC) is a commonly used stain for the detection of plant seeds [34] and microbial viability [35]. TTC is utilized to evaluate the capacity of bacteria to metabolize carbon sources by measuring their metabolic viability. The assay depends on the reduction of TTC, a colorless water-soluble compound, to insoluble TPF (triphenylformazan) by oxidoreductases and dehydrogenases [34]. The darker the color, the greater the ability to utilize the carbon source. It is well established that excessive amounts of 2,3-butanediol and TTC can have inhibitory effects on cells. Therefore, in this study, we established a gradient of final concentrations of 2,3-butanediol (5%, 1%, 0.1%, 0.05%, 0.01%, 0.005%, and 0%) and a fixed concentration of 0.005% TTC as the substrate. We then measured the activity of 2,3-butanediol dehydrogenase in the *L. lactis* N8-2 and *L. lactis* N8-1 strains. It is necessary to first centrifuge the strains during the logarithmic phase (6 h) to collect the bacterial precipitate. Subsequently, the bacteria were washed twice with IF-C and resuspended. The OD_595_ value was determined. Following this, the OD values of each sample of *L. lactis* N8-1 and *L. lactis* N8-2 strains should be adjusted to the same value. The next step involved inoculating the wells of 96-well plates with TTC and varying concentrations of 2,3-butanediol. The plates were incubated at 30 °C for 30 min, after which the results were observed and recorded by photographs.

### 2.14. Determination of the Metabolism of 2,3-Butanediol at Optimal Concentrations

The concentration of 2,3-butanediol was adjusted to 0.05%, the concentration of TTC was adjusted to 0.005%, and the other incubation conditions were the same as those described in Section 2.13. The incubation temperature was set at 30 °C, the detection wavelength was set at OD_595_ nm, and detection was performed continuously for 10 h. The metabolism of the *L. lactis* N8-1 and *L. lactis* N8-2 strains was recorded over the course of 10 h. 

### 2.15. Statistical Analysis

The data obtained are reported as the mean ± standard deviation (SD). The difference between the two groups was compared by a *t*-test with *p* < 0.05 considered significant. Statistical analyses of the data were performed using Origin 64 software (version 2024SR1).

## 3. Results and Discussion

### 3.1. Microarray Analysis of Mutants’ Phenotypes

To gain a comprehensive understanding of the strain background of these large fragment deletion strains, we employed a Biolog phenotypic microarray system. We investigated the metabolic capabilities of the deletion strains *L. lactis* N8-1 and *L. lactis* N8-2 using various carbon sources. We examined the utilization of multiple sugars, acids, and alcohols as carbon sources and collected data at 6 h, 12 h, and 18 h time points (Figure 1). In the metabolic processes involving D-maltose, sucrose, and D-fructose as carbon sources, the utilization of *L. lactis* N8-2 exhibited different levels of increase at 12 h and 18 h as opposed to the sixth hour. However, this trend was not statistically significant when compared with *L. lactis* N8-1. Correspondingly, in the metabolism of mannitol and D-arabinol, although some increase was observed in the later stages, the difference in utilization between the *L. lactis* N8-2 and *L. lactis* N8-1 strains was not found to be significant. Interestingly, *L. lactis* N8-2 demonstrated a substantial increase of approximately 3.7-fold, 4.4-fold, and 4.6-fold at 6 h, 12 h, and 18 h, respectively, compared to *L. lactis* N8-1 when 2,3-butanediol was used as the carbon source. The Biolog method is an approach used to measure the diversity of microbial metabolic functions based on the metabolic response patterns induced by different substrates. Tsigkrimani et al. [29] successfully identified lactic acid bacteria in ripened *Feta* and *Kefalograviera* cheeses by using Gen III plates combined with other molecular techniques. In another study, Connor et al. [36] tested for differences in carbon sources among three bacterial strains using PM1 assay plates. They detected significant differences in 34 carbon sources and identified multiple genes related to carbon metabolism.

### 3.2. Analysis of the Metabolic Pathways of 2,3-Butanediol and the Identification of Differentially Expressed Genes in L. lactis N8

In Biolog phenotyping, it was observed that *L. lactis* N8-2 exhibits a notably heightened ability to metabolize 2,3-butanediol. In collaboration with the *Lactococcus lactis* subsp. *lactis* IL1403 (KEGG GENOME: *Lactococcus lactis* subsp. *lactis* IL1403) metabolic pathway of *Lactococcus lactis* IL1403, we conducted a gene count for the 2,3-butanediol metabolic pathway in *L. lactis* N8-2 (Figure 2A). Each gene within this pathway plays a distinct role in the synthesis of 2,3-butanediol. First, glucose undergoes glycolysis to produce pyruvate. Pyruvate is then converted into formic acid, acetyl coenzyme A, and lactate by the *pfl*, *pdh*, and *ldh* enzymes, respectively. Additionally, pyruvate is transformed into α-acetyl lactate by the α-acetyl lactate synthetase enzyme encoded by the als gene. Afterward, α-acetyl lactate is further converted into acetoin by acetylacetate decarboxylase enzymes encoded by the *aldB* and *aldC* genes. At the same time, diacetyl can also be catalyzed by *butA* to produce acetoin. Acetoin is then utilized in the synthesis of 2,3-butanediol with the assistance of acetoin reductase/2,3-butanediol dehydrogenase encoded by the 2,3-butanediol dehydrogenase gene cluster. In this study, we explored the role of acetoin reductase/2,3-butanediol dehydrogenase in catalyzing the metabolic pathway for the generation of acetoin from 2.3-butanediol.

The biosynthesis of 2,3-butanediol in *L. lactis* N8 is enabled by a set of five crucial genes (Figure 2B). These genes differ from those found in the typical 2,3-butanediol-producing strains, *Klebsiella* sp., and *Bacillus* sp., because the key genes responsible for 2,3-butanediol synthesis are organized in a cohesive regulatory structure that is uniformly controlled by the relevant genes [37]. Nevertheless, in *L. lactis* N8, these genes are located in different regions of the genome, except *butB* and *butA*, which are transcribed together under the same promoter, P*_butBA_*. The remaining three genes formed separate transcription units that were regulated by their respective promoters (Figure 2B).

According to transcriptome sequencing analysis, in *L. lactis* N8-2, which exhibited enhanced utilization of 2,3-butanediol, the expression levels of *butA* and *butB* were approximately 32-fold and 33-fold higher, respectively, than those in *L. lactis* N8-1 (Figure 3A). Additionally, the transcript abundance of *butA* and *butB* in *L. lactis* N8-2 showed increases of approximately 24-fold and 21-fold, respectively (Figure 3B), which were slightly lower than the 30-fold difference observed in the transcriptome assay. However, the overall trend was consistent. Based on these findings, it can be concluded that the enhanced expression of the *butBA* gene in the deletion strains was primarily due to genomic alterations that occurred during the construction of *L. lactis* N8-2. Moreover, the protein encoded by this gene exhibited bi-directional catalytic activity, which, along with the characterization of its function, showed that the heightened expression of *butBA* was the primary factor contributing to the enhanced utilization capacity of 2,3-butanediol in the metabolic outcomes observed in the Biolog phenotype microarray analysis of the *L. lactis* N8-2 strain. We aimed to investigate the regulatory mechanisms of *butBA* in *L. lactis* N8-2.

### 3.3. Whole-Genome Complete Map Sequencing of L. lactis N8-2

To explore the factors contributing to the elevated expression of *butBA* in transcriptome experiments and Q-PCR results and to gain a more comprehensive understanding of the background of this strain, we conducted whole-genome sequencing of *L. lactis* N8-2. Our analysis of the complete genome sequence of *L. lactis* N8-2 revealed a single-base mutation located −46 upstream of P*_butBA_*, specifically a change from guanine to thymine, resulting in a substitution from G to T (Figure 4A). To evaluate the promoter’s expression capability following a point mutation, we initially chose appropriate reporter genes to construct plasmids and subsequently analyzed their respective expression abilities.

### 3.4. Analysis of the Promoter Region of the butBA Gene in L. lactis N8

We combined the roles of promoters in bacteria and then explored the promoter of 2,3-butanediol dehydrogenase in *L. lactis* N8. In 2008, Nieves et al. analyzed the *butBA* gene cluster of *Lactococcus lactis* IL1403 [27]. The study revealed that the two genes within the cluster, *butB* and *butA*, were transcribed from the same promoter, known as P*_butBA_*. However, the expression of these genes was weak. The promoter region −10 of P*_butBA_* was found to have a TAGAAT sequence, which differs from the regular promoter sequence TATAAT [38]. Furthermore, the spacing between the −10 and −35 regions was observed to be 14 bp, which deviates from the conventional strong promoter spacing of 16–19 bp. Based on these findings, it was concluded that the promoter of the *butBA* gene cluster could only function as a weak promoter.

### 3.5. Preliminary Testing of the Impact of Point Mutations on Promoter Activity

Based on the results of sequencing and analyses of the promoter structure, it has been determined that P*_butBA_* contains intact −10, −35, and *rbs* sequences. To validate the expression ability of the promoter, we selected the chloramphenicol (*cat*) and red fluorescent protein (*rfp*) reporter genes. In this study, we used the genomes of *L. lactis* N8-1, *L. lactis* N8-2, and pLEB124 as templates for our experiments. We successfully obtained the recombinant plasmids pLEB124-P_1_-*cat*, pLEB124-P_2_-*cat*, and pLEB124-P_45_-*cat*. Furthermore, the chloramphenicol reporter gene was substituted with *rfp* to acquire the recombinant plasmids pLEB124-P_1_-*rfp*, pLEB124-P_2_-*rfp*, and pLEB124-P_45_-*rfp* (Figure 5A).

The minimum inhibitory concentration (MIC) of the N8-1-P_45_-*cat*, N8-1-P_1_-*cat*, N8-1-P_2_-*cat*, N8-2-P_45_-*cat*, N8-2-P_1_-*cat*, and N8-2-P_2_-*cat* strains was tested against 150 μg/mL chloramphenicol. The results indicated that, with the exception of the positive control (pLEB124-P_45_-*cat*), no significant changes in tolerance were observed (Table 3). Furthermore, in the fluorescence intensity assays conducted on strains N8-1-P_1_-*rfp*, N8-1-P_2_-*rfp*, N8-2-P_1_-*rfp*, and N8-2-P_2_-*rfp*, no significant difference in fluorescence intensity was observed.

### 3.6. Structural Analysis of the Transcriptionally Active Functional Region of the butBA Gene Cluster

Through previous promoter validation studies, we could not determine the cause of the increased expression of the *butBA* gene. Therefore, we tested the expression ability of the P*_butBA_* promoter by linking the complete *butB* gene post-transcription. pLEB124-P_1_-*butB*-*cat* and pLEB124-P_2_-*butB*-*cat* were constructed following the methods described in Figure 5B. N8-1-P_1_-*butB*-*cat*, N8-1-P_2_-*butB*-*cat*, N8-2-P_1_-*butB*-*cat*, and N8-2-P_2_-*butB*-*cat* were analyzed for chloramphenicol tolerance assays, as shown in Table 4. The strain harboring plasmid pLEB124-P_2_-*butB*-*cat*, which carries a mutated promoter, exhibited 11-fold higher tolerance to chloramphenicol than the strain with the unmutated promoter. This observation implies that a point mutation in the P*_butBA_* promoter significantly affects its transcriptional expression. It can be inferred that this mutation is located within the binding site of an unidentified transcriptional repressor in *L. lactis* N8-2.

### 3.7. Mechanistic Analysis and Exploration of butBA Promoter Point Mutation Resulting in Enhanced Expression of Its Gene Cluster

In the above experiment, we comprehensively analyzed and verified the underlying cause of elevated expression of the *butBA* gene cluster in *L. lactis* N8-1 and *L. lactis* N8-2 strains. We identified a single-base mutation at position −46 in the P*_butBA_* promoter of *L. lactis* N8-2, specifically a mutation from guanine to thymine, as the factor responsible for this observed phenotype. Additionally, our sequencing results indicated no alterations in the −35 region, −10 region, or spacer region between the two. Previous experiments have established that the primary factor contributing to the increased transcriptional ability of the P*_butBA_* promoter is the occurrence of point mutations. RegPrecise (http://regprecise.lbl.gov, accessed on 1 September 2021) is a web resource for the collection, visualization, and analysis of transcriptional regulons reconstructed by comparative genomics [39]. By consulting this database, we find that regulation of *butB* and *butA* by the global transcriptional regulator CcpA in *Lactococcus lactis* IL1403 has been included in this website. The binding motifs of CcpA for both *butB* and *butA* were identified with a score of 4.9, and the binding sequence was determined to be AAGAAAACGCTTTAAA (Figure 4B). Notably, this sequence is identical to the *cre* target site of the CcpA protein in *L. lactis* N8.

A literature review revealed that the homologous region of 16 bp in the P*_butBA_* promoter shares an identical sequence with *cre*, which serves as the target site for the CcpA protein. The CcpA protein is known to regulate carbon metabolism in *Lactococcus lactis* [40]. In a study, Aldert L. Zomer et al. [40] noted significant changes in the transcription of numerous genes related to carbon source metabolism after the deletion of CcpA proteins in *Lactococcus lactis* MG1363. In particular, a 3- to 5-fold increase in *butBA* gene cluster transcription was observed. Conversely, our findings demonstrate a much higher transcription initiation ability following a single-base mutation in the P*_butBA_* promoter in *L. lactis* N8-2.

### 3.8. Preparation of Fluorescent Probes and Purification of CcpA Proteins

The genomes of *L. lactis* N8-1 and N8-2 were used as templates for this study. PCR amplification was performed using primers containing homology arm sequences of the 16 bp cy5 primer sequence to obtain the target fragments N8-1-*cre* (97 bp), N8-2-*cre* (97 bp), and N8-16s (266 bp) (Figure 6A). The ligation vector pEASY-T1 was used to generate pEASY-N8-1-*cre*, pEASY-N8-2-*cre*, and pEASY-N8-16s recombinant vectors through white/blue screening. Each of the three recombinant plasmids served as a template for subsequent PCR amplification using the cy5 fluorescent primer (Figure 6B). Following the heterologous expression of pETm3c-CcpA, protein purification was conducted, and the eluted target proteins were desalted using dextran gel G-75. SDS-PAGE results were collected for further analysis (Figure 6C).

### 3.9. Micro-Scale Thermophoresis (MST)

To determine the KD value of the apparent kinetic constant for the binding of the CcpA protein to the nucleic acid molecules N8-1-*cre*, N8-2-*cre*, and N8-16s and to investigate the binding of CcpA to N8-1-*cre* and N8-2-*cre*, with N8-16s as a control, we compared the binding kinetic curves and affinity constants. This was performed by monitoring the distribution of fluorescent signals and measuring the molecular distribution in the temperature gradient field. MST experiments were performed to determine the apparent kinetic constant KD for protein binding to nucleic acid molecules. The concentration-stationary phase utilized Cy5 probe-labeled DNA, whereas the target protein was gradient-diluted using a 2-fold dilution method. The resulting binding kinetic curves of CcpA and the target DNA were analyzed. Finally, we obtained binding kinetic curves and affinity constants. The binding of the target protein to N8-1-*cre* can be accurately represented by an “S-shaped” curve, indicating a strong binding trend between the two. The measured affinity constant K_d_ was 0.044 ± 0.067 μM (Figure 7A). Nevertheless, when the binding motif (*cre*) was mutated, the binding of the global transcriptional regulator CcpA to N8-2-*cre* was disorganized and scattered. In this case, an “S-shaped” curve could not be fitted, suggesting the absence of a binding trend between the target protein and mutated motif. Consequently, the affinity constant cannot be determined (Figure 7B). At the same time, the interaction between CcpA and the 16s DNA showed a disordered scattering pattern, which made it impossible to measure the affinity constant (Figure 7C). When there is a clear binding trend between the binding protein and the target DNA, the results exhibit an “S-shaped” curve [41]. Conversely, in the absence of a binding trend between the binding protein and the target DNA, the results appear as scattered dots. The MST assay showed that a mutation in the binding motif resulted in the loss of CcpA-binding capability and its subsequent inability to interact with the binding motif. The DNA-binding properties of CcpA are key to its global function, which includes the utilization of two separable DNA-binding modules, the HTHLH motifs and the hinge helices. When the *cre* motif is changed, the binding of CcpA to *cre* is also changed. The binding curves were randomly scattered, signifying that the mutation caused CcpA, which is the regulator of the 2.3-butanediol dehydrogenase gene cluster in *L. lactis* N8-2, to detach from *cre* and lose its role in regulating the gene cluster. This resulted in the activation of the expression of downstream genes.

### 3.10. Construction of a Bacterial System for Fluorescent Hybridization Using Two Plasmids

To study the in vivo binding of CcpA to the *cre* site, we generated target validation strains by constructing N8-1-*cre* and N8-2-*cre* DNA sequences at the N-terminus of the eGFP gene in the pACYC-eGFP vector. Moreover, the CcpA gene was constructed using the pRSF-Duet vector (Figure 8A). Electrotransformation of pRSF-CcpA with pACYC-N8-1-*cre*-eGFP and pACYC-N8-2-*cre*-eGFP was performed in *E. coli* BL21(DE3) competent cells (Figure 8B).

### 3.11. Determination of Growth Trends and Fluorescence Values in Two-Hybrid Plasmid Strains

To provide further evidence of CcpA binding and dissociation from the *cre* site in vivo, we used a fluorescent reporter system to visualize the fluorescent strains. (Figure 8C). The fluorescence values of the strains carrying only the N8-1-*cre*-eGFP and N8-2-*cre*-eGFP plasmids were similar. However, when CcpA was present, the expression of green fluorescent protein in the CcpA-N8-1-*cre*-eGFP hybrid strain was significantly reduced. In contrast, there was no significant change in the intensity of green fluorescent protein expression in the CcpA-N8-2-*cre*-eGFP strain initiated by N8-2-*cre*. Therefore, it can be concluded that CcpA did not bind to *cre*. As a result, downstream eGFP gene transcription was not affected, and there was no significant change in green fluorescent protein expression. The results presented above indicate that the mutation of guanine to thymine at the eighth position in the *cre* sequence 5′-AAGAAAACGCTTTAAA-3′ is crucial for the alleviation of the transcriptional repression of downstream genes by CcpA. 

### 3.12. A Comparative Analysis of the Metabolic Responses of L. lactis N8-2 and L. lactis N8-1 to Varying Concentrations of 2,3-Butanediol

According to the findings from the Biolog phenotype microarray analysis, it was observed that the utilization rate of *L. lactis* N8-2 could be enhanced four times compared to that of *L. lactis* N8-1 when 2,3-butanediol was employed as the carbon source. To compare the catalytic activity of 2,3-butanediol by 2,3-butanediol dehydrogenase in *L. lactis* N8-1 and *L. lactis* N8-2, and thus the utilization of the substrate by both strains, an investigation was conducted to determine the optimal concentration of 2,3-butanediol metabolized by *L. lactis* N8-2 in the presence of 2,3,5-triphenyl-2H-tetrazolium chloride (TTC) as an indicator. Tetrazolium salts, which are colorless compounds, become colored when reduced to form formazans. They are used as indicators in prokaryotic cells to detect dehydrogenase activity in metabolism [42]. The reduced form of TTC has a distinctive color that enables the accurate and easy detection of enzyme metabolism. The metabolic viability of the strains was assessed. The gradient concentrations of 2.3-butanediol are shown in Figure 9A. Based on the results shown in Figure 9B, the most significant change in color was observed at a substrate concentration of 0.01%, indicating that *L. lactis* N8-2 exhibited the highest metabolic rate at this concentration. According to the results, 2,3-butanediol dehydrogenase was more catalytically active in metabolizing 2,3-butanediol in *L. lactis* N8-2 than in *L. lactis* N8-1. This is because, in the point mutant strain *L. lactis* N8-2, the mutation of *cre* causes dissociation of CcpA and *cre*. Subsequently, in the regulation of central carbon metabolism, the global transcriptional regulator CcpA deregulates the repression of the 2,3-butanediol dehydrogenase gene cluster and activates its expression.

To provide a clearer understanding of the variations in the utilization of 2,3-butanediol by the two strains, we analyzed the absorbance over the initial ten-hour period at concentrations of 0.005% TTC and 0.01% 2,3-butanediol. As depicted in Figure 9C, the catalytic activity of the *L. lactis* N8-2 metabolism exhibited a continuous increase, indicating the high catalytic activity of 2,3-butanediol dehydrogenase after derepression of CcpA inhibition. In contrast, *L. lactis* N8-1 is regulated by inhibition of CcpA, resulting in a very low metabolic capacity when 2.3-butanediol is used as a carbon source.

Bacterial transcription factors are typically composed of a regulatory domain and a DNA-binding domain. The DNA-binding domain binds to the bacterial promoter region or transcriptional region, either repressing or activating transcription [43]. Studies have shown that the global transcriptional regulators CodY and CcpA in *Staphylococcus aureus* play key roles in central carbon metabolism, virulence gene expression, biofilm formation, and the preferential use of fast-acting carbon sources [44]. Positive and negative regulation of the transcription of CcpA-regulated genes involves the binding of CcpA to acting *cre* [45]. The expression of genes associated with pyruvate metabolism is enhanced by CcpA inactivation in *Lactobacillus delbrueckii* subsp. *Bulgaricus* [10].

Metabolic engineering of transcription factors has proven to be an effective strategy to increase the yield of target products [46,47,48]. Nevertheless, other carbon sources can also be metabolized by altering the direction of transcription factor regulation through molecular biology. CcpA is associated with the serine-phosphorylated form of the phosphocarrier protein HPr (P-ser-HPr), which binds to *cre* and is typically found near the promoter. This interaction can either repress or enhance the transcription of downstream operons [8]. Based on the regulatory mechanism of CcpA, coupled with the results obtained from our experimental validation, we mapped the regulation of the 2,3-butanediol dehydrogenase gene cluster by CcpA in *L. lactis* N8 (Figure 10). In *L. lactis* N8-1, the *cre* gene is not mutated, and CcpA can bind to *cre* with the help of P-ser-HPr and RNA polymerase. This binding causes a negative regulatory effect on downstream genes, leading to repression of the transcriptional expression of the 2,3-butanediol dehydrogenase gene cluster (Figure 10A). In *L. lactis* N8-2, the CcpA protein no longer binds to the *cre* site due to a mutation (Figure 10B). Consequently, downstream genes are activated, functioning as a switch to release repression and enable high expression of the 2,3-butanediol dehydrogenase gene cluster. This activation provides scope for the utilization of 2,3-butanediol. The 2,3-butanediol dehydrogenase, which was liberated from the suppression of CcpA carbon metabolism in *L. lactis* N8-2, exhibited a more robust catalytic capability. It is crucial to culture and grow bacteria in the laboratory, and point mutations in the genome during bacterial replication can occasionally result in unforeseen phenotypic effects. While most previous reports in the literature have focused on the effects of deleting the CcpA gene on bacterial physiological metabolism, our study focuses on the regulation of downstream genes through mutation of the DNA-binding structural domain.

To conclude, the present study found that mutations in *cre* deregulate the negative regulatory effect of the global transcriptional regulator CcpA on the 2,3-butanediol dehydrogenase gene cluster. This reveals a critical role for CcpA in the regulation of central carbon metabolism and provides biological evidence for the bacterial enhancement of the catalytic potency of 2,3-butanediol dehydrogenase.

## Figures and Tables

**Figure 1 microorganisms-12-00773-f001:**
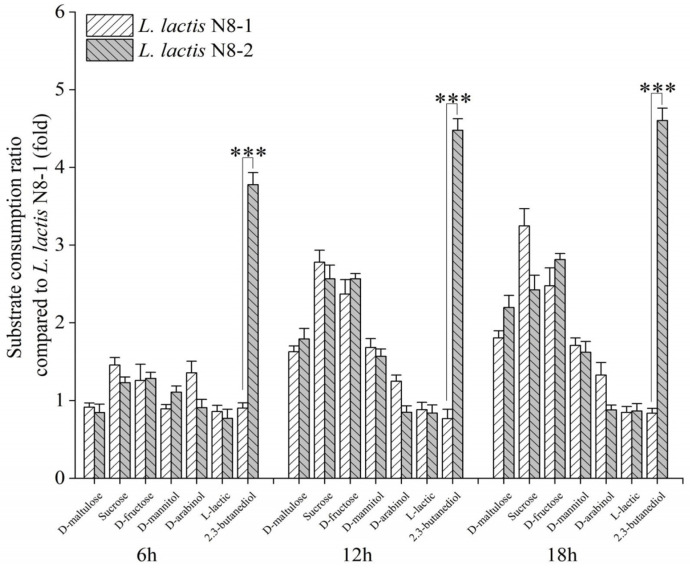
Detection of the metabolic capabilities of *L. lactis* N8-1 and *L. lactis* N8-2 genome-deleted chassis strains toward various carbon sources. *** *p* < 0.001 (significant difference between the mutant and control).

**Figure 2 microorganisms-12-00773-f002:**
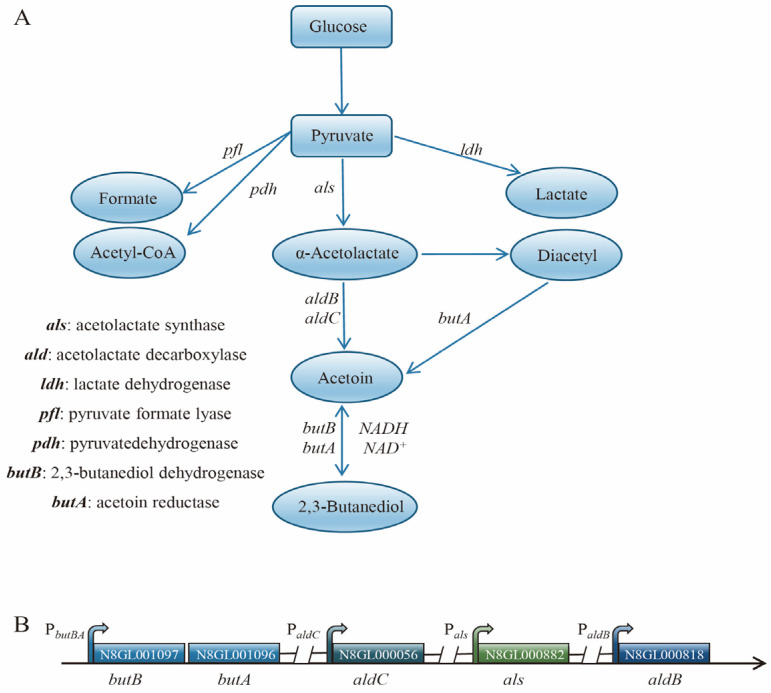
Metabolic pathway of 2,3-butanediol in *L. lactis* N8 and statistical analysis of associated genes. (**A**) 2,3-butanediol metabolic pathway and the functions of related genes. (**B**) Localization and transcriptional unit analysis of key genes implicated in the metabolism of 2,3-butanediol.

**Figure 3 microorganisms-12-00773-f003:**
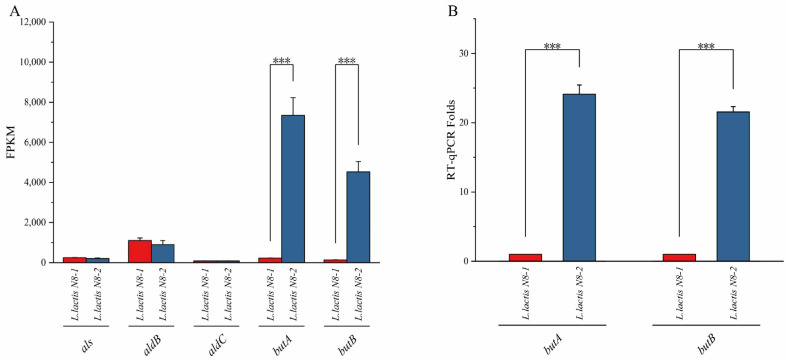
(**A**) Transcriptome sequencing analysis of genes associated with the metabolic pathway of 2,3-butanediol in *L. lactis* N8-1 and *L. lactis* N8-2. (**B**) Transcriptional alterations in the metabolic pathway of 2,3-butanediol in *L. lactis* N8-1 and *L. lactis* N8-2. *** *p* < 0.001 (significant difference between the mutant and control).

**Figure 4 microorganisms-12-00773-f004:**
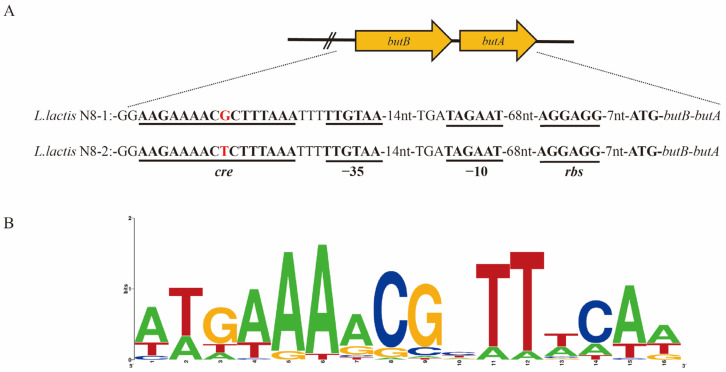
Identification of a single mutant base in *L. lactis* N8-2 and binding motifs of CcpA-regulated genes in *Lactococcus lactis* IL1403. (**A**) Results of sequence difference comparison between *L. lactis* N8-2 whole genome sequencing and *L. lactis* N8-1 upstream of the 2,3-butanediol dehydrogenase gene promoter. (**B**) Base sequences of all binding sites regulated by CcpA transcription factors in *Lactococcus lactis* IL1403 were accessed in the RegPrecise web resource.

**Figure 5 microorganisms-12-00773-f005:**
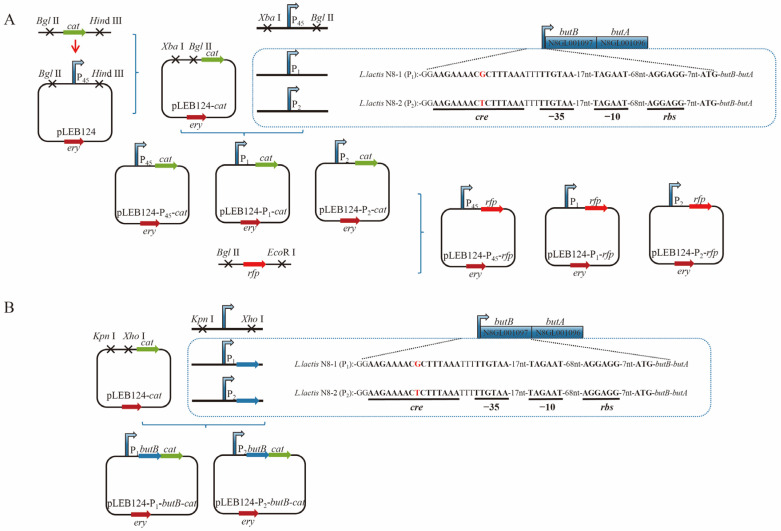
Validation of P_1_, P_2_, and P_45_ promoter viability in the pLEB124 vector using *cat* and *rfp* as reporter genes. (**A**) Verification of the viability of the *butBA* gene promoters P_1_ and P_2_ in *L. lactis* N8-1 and *L. lactis* N8-2 using *cat* and *rfp* as reporter genes in the pLEB124 vector, while setting the P_45_ promoter as a positive control. (**B**) Validation of the viability of the P_1_-*butB* and P_2_-*butB* promoters linked to the *butB* gene using *cat* as a reporter gene in pLEB124.

**Figure 6 microorganisms-12-00773-f006:**
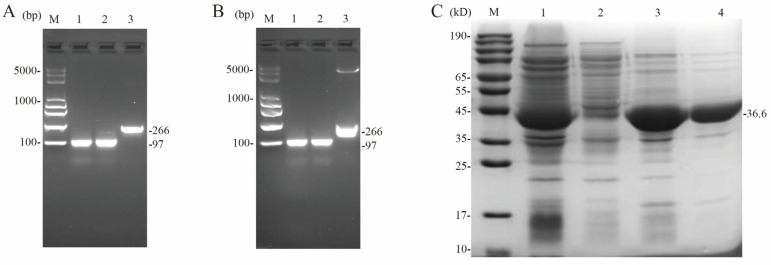
Preparation of DNA probes with CcpA protein in MST experiments. (**A**) *L. lactis* N8-1 and *L. lactis* N8-2 were used as templates for the first amplification with primers carrying the Cy5 sequence target. Lane M: DNA Marker; lane 1: N8-1-*cre* (97 bp); lane 2: N8-2-*cre* (97 bp); lane 3: N8-16s (266 bp); (**B**) Secondary amplification with pEASY-N8-1-*cre*, pEASY-N8-2-*cre*, and pEASY-N8-16s as template amplification. M: DNA Marker; lane 1: N8-1-*cre* (97 bp); lane 2: N8-2-*cre* (97 bp); lane 3: N8-16s (266 bp); (**C**) Purification outcomes of target proteins through the heterologous expression of pETm3c-CcpA. Lane M: Protein Marker; lane 1: whole bacterial proteins collected after cell fragmentation; lane 2: elution of uncolumnized proteins; lane 3 and lane 4: target proteins collected after 0.3 M imidazole elution.

**Figure 7 microorganisms-12-00773-f007:**
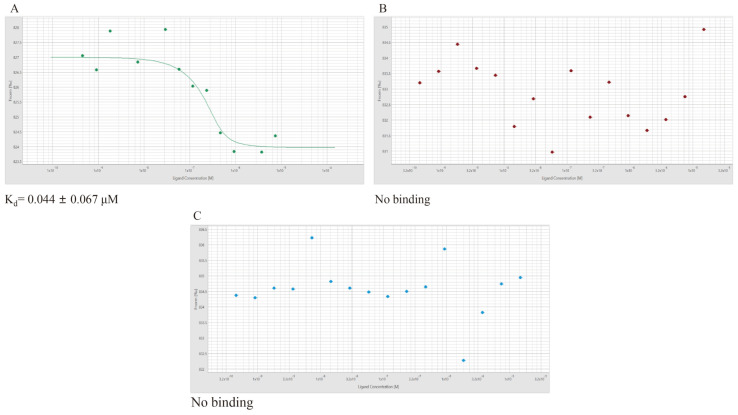
(**A**) The binding of the CcpA protein to fluorescently labeled N8-1-*cre* was analyzed using micro-scale thermophoresis (MST). Changes in the thermophoretic signal resulted in binding affinities of 0.044 ± 0.067 μM. (**B**) Measurement of the interaction between the CcpA protein and fluorescently labeled N8-2-*cre* using MST showed no binding under the tested conditions. (**C**) Measurement of the interaction between the CcpA protein and fluorescently labeled N8-16s using MST showed no binding under the tested conditions.

**Figure 8 microorganisms-12-00773-f008:**
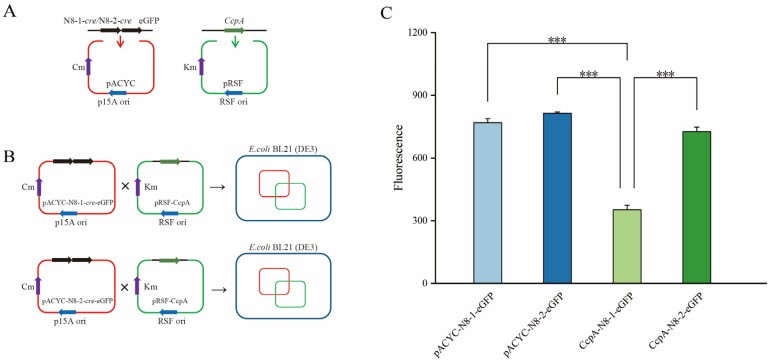
Construction of a two-plasmid fluorescent hybridization system in *Escherichia coli* BL21(DE3). (**A**) Construction of pACYC-N8-1-*cre*-eGFP, pACYC-N8-2-*cre*-eGFP, and pRSF-CcpA plasmids. (**B**) Plasmids were combined with *E. coli* BL21 (DE3) cells. (**C**) Determination of fluorescence values for the four bacterial strains. *** *p* < 0.001 (significant difference between the mutant and control).

**Figure 9 microorganisms-12-00773-f009:**
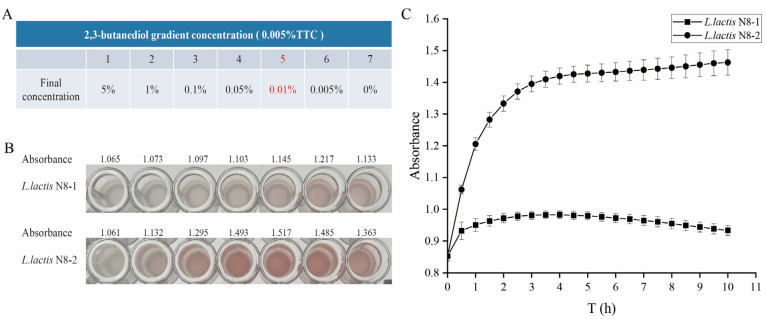
Assay of the ability of *L. lactis* N8-1 and *L. lactis* N8-2 to metabolize 2,3-butanediol. (**A**) Setting up a gradient concentration of 2,3-butanediol (5%, 1%, 0.1%, 0.05%, 0.01%, 0.005%, and 0%) and the final concentration of 0.005% TTC (The red mark of 0.01% is the optimal concentration). (**B**) Color response of *L. lactis* N8-1 and *L. lactis* N8-2 to metabolize 2,3-butanediol. (**C**) Detection of *L. lactis* N8-1 and *L. lactis* N8-2 metabolism at optimal 2,3-butanediol concentrations.

**Figure 10 microorganisms-12-00773-f010:**
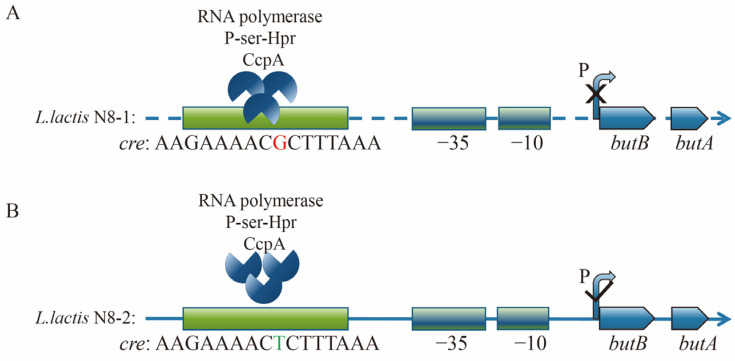
Schematic representation of the mechanism whereby CcpA regulates the 2,3-butanediol dehydrogenase gene cluster in *L. lactis* N8-1 and *L. lactis* N8-2. (**A**) Schematic representation of transcriptional regulation of the 2,3-butanediol dehydrogenase gene cluster downstream of CcpA-binding *cre* repression in *L. lactis* N8-1. (**B**) Schematic representation of the inhibitory regulatory effect of CcpA dissociation and *cre* loss on the downstream 2,3-butanediol dehydrogenase gene cluster in *L. lactis* N8-2.

**Table 1 microorganisms-12-00773-t001:** Bacterial strains and plasmids utilized in this study.

Strains or Plasmids	Relevant Descriptions	Reference
Strains		
*L. lactis* N8	Wild-type (WT) Nisin Z producer	[28]
*L. lactis* N8-1	The first DNA region L1 deletion in *L. lactis* N8	[6]
*L. lactis* N8-2	The L2 deletion in *L. lactis* N8-1	Lab stork
*E. coli* DH5α	Cloning host; F-φ80 lacZΔM15 endA1 recA1 endA1 hsdR17 (rK-mK+) supE44 thi-1 gyrA 96 relA1 Δ(lacZYA-argF)U169 deoR λ-	Lab stork
BL21(DE3)	F-ompT hsdSB(rB-m-) gal dcm (DE3)	Lab stork
N8-1-P_45_-*cat*	Em^r^, *L. lactis* N8-1 derivative containing pLEB124-P_45_-*cat*	This study
N8-1-P_1_-*cat*	Em^r^, *L. lactis* N8-1 derivative containing pLEB124-P_1_-*cat*	This study
N8-1-P_2_-*cat*	Em^r^, *L. lactis* N8-1 derivative containing pLEB124-P_2_-*cat*	This study
N8-2-P_45_-*cat*	Em^r^, *L. lactis* N8-2 derivative containing pLEB124-P45-*cat*	This study
N8-2-P_1_-*cat*	Em^r^, *L. lactis* N8-2 derivative containing pLEB124-P_1_-*cat*	This study
N8-2-P_2_-*cat*	Em^r^, *L. lactis* N8-2 derivative containing pLEB124-P_2_-*cat*	This study
N8-1-P_45_-*rfp*	Em^r^, *L. lactis* N8-1 derivative containing pLEB124-P_45_-*rfp*	This study
N8-1-P_1_-*rfp*	Em^r^, *L. lactis* N8-1 derivative containing pLEB124-P_1_-*rfp*	This study
N8-1-P_2_-*rfp*	Em^r^, *L. lactis* N8-1 derivative containing pLEB124-P_2_-*rfp*	This study
N8-2-P_45_-*rfp*	Em^r^, *L. lactis* N8-2 derivative containing pLEB124-P_45_-*rfp*	This study
N8-2-P_1_-*rfp*	Em^r^, *L. lactis* N8-2 derivative containing pLEB124-P_1_-*rfp*	This study
N8-2-P_2_-*rfp*	Em^r^, *L. lactis* N8-2 derivative containing pLEB124-P_2_-*rfp*	This study
N8-1-P_1_-*butB*-*cat*	Em^r^, *L. lactis* N8-1 derivative containing pLEB124-P_1_-*butB*-*cat*	This study
N8-1-P_2_-*butB*-*cat*	Em^r^, *L. lactis* N8-1 derivative containing pLEB124-P_2_-*butB*-*cat*	This study
N8-2-P_1_-*butB-cat*	Em^r^, *L. lactis* N8-2 derivative containing pLEB124-P_1_-*butB-cat*	This study
N8-2-P_2_-*butB-cat*	Em^r^, *L. lactis* N8-2 derivative containing pLEB124-P_1_-*butB-cat*	This study
pACYC-N8-1-*cre-rfp*	Cm^r^, *E. coli* BL21 containing pACYC-N8-1-*cre-rfp*	This study
pACYC-N8-2-*cre-rfp*	Cm^r^, *E. coli* BL21 containing pACYC-N8-2-*cre-rfp*	This study
CcpA-N8-1-*cre*-eGFP	Kan^r^, Cm^r^, *E. coli* BL21containing pRSF-CcpA and pACYC-N8-1-*cre*-*rfp*	This study
CcpA-N8-2-*cre*-eGFP	Kan^r^, Cm^r^, *E. coli* BL21containing pRSF-CcpA and pACYC-N8-2-*cre*-*rfp*	This study
Plasmids		
pEASY-T1	Amp^r^, Blue-white screening vector	TransGen, Beijing, China
pLEB124	Em^r^, expression vector	[28]
pETm3c	Amp^r^, expression vector	Lab stork
pNZ8048-*rfp*	Cm^r^, pNZ8048 derivative containing the *rfp* gene	Lab stork
pRSF-Dute	Kan^r^, pRSF-Duet is designed for the coexpression of two target ORFs.	Lab stork
pACYC-eGFP	Cm^r^, pACYC-eGFP is designed for the coexpression of two target genes.	Lab stork
pETm3c-*CcpA*	Amp^r^, pETm3c derivative containing the *CcpA* gene	This study
pEASY-N8-1-*cre*	Amp^r^, pEASY-T1 derivative containing the N8-1-*cre* gene	This study
pEASY-N8-2-*cre*	Amp^r^, pEASY-T1 derivative containing the N8-2-*cre* gene	This study
pEASY-N8-16s	Amp^r^, pEASY-T1 derivative containing the N8-16s gene	This study
pACYC-N8-1-*cre*-eGFP	Cm^r^, pACYC-eGFP derivative containing the N8-1-*cre* gene	This study
pACYC-N8-2-*cre*-eGFP	Cm^r^, pACYC-eGFP derivative containing the N8-2-*cre* gene	This study
pRSF-*CcpA*	Kan^r^, pRSF-Dute derivative containing the *CcpA* gene	This study
pLEB124-P_45_-*cat*	Em^r^, pLEB124 derivative containing the P_45_ and *cat* cassette	This study
pLEB124-P_1_-*cat*	Em^r^, pLEB124 derivative containing the P_1_ and *cat* cassette	This study
pLEB124-P_2_-*cat*	Em^r^, pLEB124 derivative containing the P_2_ and *cat* cassette	This study
pLEB124-P_45_-*rfp*	Em^r^, pLEB124 derivative containing the P_45_ and *rfp* cassette	This study
pLEB124-P_1_-*rfp*	Em^r^, pLEB124 derivative containing the P_1_ and *rfp* cassette	This study
pLEB124-P_2_-*rfp*	Em^r^, pLEB124 derivative containing the P_2_ and *rfp* cassette	This study
pLEB124-P_1_-*butB*-*cat*	Em^r^, pLEB124 derivative containing the P_1_ and *butB* and *cat* cassette	This study
pLEB124-P_2_-*butB*-*cat*	Em^r^, pLEB124 derivative containing the P_2_ and *butB* and *cat* cassette	This study

**Table 2 microorganisms-12-00773-t002:** Primers in this study.

Primer	Primer Sequence (5′-3′)
*cat*-clone-F	GCTAGATCTAGGCATATCAAATGAA
*cat*-clone-R	GCTAAGCTTCCAATCATCAATTGCG
P_45_-F	GCTTCTAGACGTTAGGGGCTTGAAC
P_45_-R	GCTAGATCTTTTTCACTATTCTAGGTTCC
*rfp*-clone-F	CTAGATCTCTCGAGATGGTTTCAAAAGGTGAAG
*rfp*-clone-R	GCCGAATTCCAATTGAACGTTTCAAGCC
P*_butBA_*-F	GCTTCTAGACAATTAATAAGAATACAAAAAGTAA
P*_butBA_*-R	GCTGCTAGATCT AAAAACGCCTCCTCTAT
P*_butBA_*-*butB*-F	GCTGGTACCCAATTAATAAGAATACAAAAAGTAA
P*_butBA_*-*butB*-R	GCTCTCGAGTTTATAGACCTTTACCAGTTG
*CcpA*-F	CGGGATCCATGGTAGAATCAACAACAACAAT
*CcpA*-R	CGAGCTCTTTGGTAGAACGACGAGAAAAGA
Cy5′	AGCCAGTGGCGATAAG
N8-1/N8-2-*cre*-F	AGCCAGTGGCGATAAGTACGATAAAATATTTAAAAGTCT
N8-1/N8-2-*cre*-R	CTATCATTATTTCTAATAATTTAC
16s-F	AGCCAGTGGCGATAAGGCGTTAGCTGCGATACAG
16s-R	CATGTGTAGCGGTGAAATG
N8-1-*cre-rfp*-F	CGGGATCCCCAAGGAAGAAAACGCTTTAAATTTTTATGGTGAGCAAGGGCGAGGAGG
N8-2-*cre-rfp*-F	CGGGATCCCCAAGGAAGAAAACTCTTTAAATTTTTATGGTGAGCAAGGGCGAGGAGG
For the RT-qPCR	
Q-*tufA*-F	GACCTCTTGAGCGAATACGACT
Q-*tufA*-R	TTCTTCAACTTTAGCAACCCATT
Q-*butA*-F	ATTATCAACGCAACCTCACAAGC
Q-*butA*-R	TCCCCATTCATCATCTTTACCAG
Q-*butB*-F	CCGTCAGCAGAACATCCTAATC
Q-*butB*-R	TCAGCCAATCCTCCACCAT
T7-F	TAATACGACTCACTATAGGG
T7-R	TGCTAGTTATTGCTCAGCGG

**Table 3 microorganisms-12-00773-t003:** Verification of the impact of downstream structural genes on promoter transcription of the P*_butBA_* promoter.

Recombinant Plasmid	Host Bacterium MIC (μg/mL)	
	*L. lactis* N8-1	*L. lactis* N8-2
pLEB124-P_45_*-cat*	10	10
pLEB124-P_1_-*cat*	2.5	2.5
pLEB124-P_2_-*cat*	2.5	2.5

**Table 4 microorganisms-12-00773-t004:** Validation of the viability of the P_1_-*butB* and P_2_-*butB* promoters linked to the *butB* gene.

Recombinant Plasmid	Host Bacterium MIC (μg/mL)	
	*L. lactis* N8-1	*L. lactis* N8-2
pLEB124-P_45_-*cat*	10	10
pLEB124-P_1_-*butB*-*cat*	7.5	7.5
pLEB124-P_2_-*butB*-*cat*	80	80

## Data Availability

Data are contained within the article.

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
