# Peer review of "A Point Mutation in Cassette Relieves the Repression Regulation of CcpA Resulting in an Increase in the Degradation of 2,3-Butanediol in Lactococcus lactis"

_microorganisms, 2024, doi:10.3390/microorganisms12040773_

Round 1

Reviewer 1 Report

Comments and Suggestions for Authors

In this manuscript, the authors characterize a new L. lactis strain that contains a point mutation in the CcpA binding site. This mutation results in the increment in the degradation of 2,3 butanediol changing the metabolism of 4-carbon compounds. Interestingly, the authors present a solid work describing a transcriptional de-regulation mechanism where the CcpA transcription factor participates. However, it is necessary that the authors of this work can resolve some aspects of the manuscript.

1.        The legends of the figures must be self-explained containing the most relevant information. However, several of them are not according to an accurate legend of figure, especially legends of figures 4, 5, 6, and 9. Please, review every legend in the manuscript and correct it.

2.        Results associated with Figure 6 are not explained adequately. First, the rationale for the amplification described is not completely clear. Second, the description of the legend of the figure must be improved, because we do not know what is loaded in each line of the gels shown in the figure. 

3.        The effect of only one point mutation in the CcpA binding site of the promoter enormously affects the carbon metabolism associated with a deregulation of the negative effect of CcpA on the expression of “but” genes of this strain. However, I suggest that the authors add a little paragraph discussing how this mutation may affect the ability of CcpA to avoid binding to the promoter. How only one change in the binding site is associated with this effect on CcpA function. Maybe the author could relate these comments with the results presented in section 2.9.

4.        In Figure 9, the legend of the figure does not have the title. 

5.        The experiments detailed in section 3.10 must be improved to a better understanding of the results presented in section 2.9 where Kd are determined and presented in a molar concentration. It is necessary to inform the molar concentration of proteins (nM for example). The same goes for DNA probes.

Reviewer 2 Report

Comments and Suggestions for Authors

In their manuscript, the Authors presented a genetic modification in Lactococcus lactis, resulting in the degradation of 2,3-Butanediol. The work is extensive and interesting, but some fragments are described too briefly and there is no broader discussion of the results. It is possible that this is due to the few scientific works available.

Minor comments below

This provides new perspectives and strategies for metabolizing 4-carbon compounds using bacteria in synthetic biology. - In the opinion of the Reviewer, it is difficult to draw broad conclusions regarding the entire group of 4-carbon compounds based on the use of the decomposition of one compound.

“Many microorganisms, such as Klebsiella, Bacillus, Enterobacter, Saccharomyces, Serratia” – This is a gross simplification. In the Reviewer's opinion, one should write either - Many microorganisms, such as Klebsiella sp. etc. ... or "Many microorganisms of the genus Klebsiella etc..."

In the Reviewer's opinion, the numerical values on the Y axis in Figure 1 will be more legible.

Why didn't the Authors test the simplest source of carbon, which is glucose?

In the Reviewer's opinion, Figures 2, 5, 7 are too small and therefore not legible.

In the Reviewer's opinion, in order to better visualize the effects, the data from Figure 9 B should be presented in the supplementary materials.

In contrast, L. lactis N8-1 had very low catalytic activity, suggesting that it is regulated by CcpA in an inhibitory manner - In the opinion of the Reviewer, based on the presented results, this conclusion of the Authors is too general.
